# Load-based divergence in the dynamic allostery of two TCRs recognizing the same pMHC

Ana Cristina Chang-Gonzalez[1], Aoi Akitsu[2,3,4], Robert J Mallis[2,3,5], Matthew J Lang[6,7], Ellis L Reinherz[2,3,4], Wonmuk Hwang[1,8,9,10]*

[1]Department of Biomedical Engineering, Texas A&M University, College Station, United States; [2]Laboratory of Immunobiology, Dana-Farber Cancer Institute, Boston, United States; [3]Department of Medical Oncology, Dana-Farber Cancer Institute, Boston, United States; [4]Department of Medicine, Harvard Medical School, Boston, United States; [5]Department of Dermatology, Harvard Medical School, Boston, United States; [6]Department of Chemical and Biomolecular Engineering, Vanderbilt University, Nashville, United States; [7]Department of Molecular Physiology and Biophysics, Vanderbilt University, Nashville, United States; [8]Department of Materials Science & Engineering, Texas A&M University, College Station, United States; [9]Center for AI and Natural Sciences, Korea Institute for Advanced Study, Seoul, Republic of Korea; [10]Department of Physics & Astronomy, Texas A&M University, College Station, United States

*For correspondence:
hwm@tamu.edu

Competing interest: The authors declare that no competing interests exist.

## eLife Assessment

This **useful** study reports detailed molecular dynamics (MD) simulations of T-cell receptors (TCRs) in complex with a peptide/MHC complex, for a better understanding of the mechanism of T-cell activation. The MD simulations provide **solid** evidence supporting that different TCRs can respond mechanically in different ways upon binding to the same pMHC complex. The analyses are systematic and provide testable predictions that can be evaluated by future mutagenesis and force microscopy studies.

**Abstract** Increasing evidence suggests that mechanical load on the αβ T-cell receptor (TCR) is crucial for recognizing the antigenic peptide-bound major histocompatibility complex (pMHC) molecule. Our recent all-atom molecular dynamics (MD) simulations revealed that the inter-domain motion of the TCR is responsible for the load-induced catch bond behavior of the TCR-pMHC complex and peptide discrimination (Chang-Gonzalez et al., 2024). To further examine the generality of the mechanism, we perform all-atom MD simulations of the B7 TCR under different conditions for comparison with our previous simulations of the A6 TCR. The two TCRs recognize the same pMHC and have similar interfaces with pMHC in crystal structures. We find that the B7 TCR-pMHC interface stabilizes under ~15 pN load using a conserved dynamic allostery mechanism that involves the asymmetric motion of the TCR chassis. However, despite forming comparable contacts with pMHC as A6 in the crystal structure, B7 has fewer high-occupancy contacts with pMHC and exhibits higher mechanical compliance during the simulation. These results indicate that the dynamic allostery common to the TCRαβ chassis can amplify slight differences in interfacial contacts into distinctive mechanical responses and nuanced biological outcomes.

## Introduction

The A6 T-cell receptor (TCR)αβ and B7 TCRαβ (herein we call TCRαβ simply as TCR) are both specific for the same Tax peptide (LLFGYPVYV) of the human T lymphotropic virus 1 (HTLV-1) bound to HLA-A2 (*Garboczi et al., 1996a*, *Garboczi et al., 1996b*; *Ding et al., 1998*). A6 and B7 derive from T-cell clones isolated from two patients with HTLV-1-associated myelopathic/tropical spastic paraparesis (*Utz et al., 1996*; *Ding et al., 1998*). They share the same Vβ germline gene (TRBV6-5) and differ only in the Vα germline gene (A6: TRAV12-2; B7:TRAV29/DV5), with sequence similarity of 45% for Vα, 96% for Vβ, and 100% for Cα and Cβ (*Ding et al., 1998*). The only structural differences between the two TCRs are from the residues of the Vα domain and the highly variable complementarity-determining region 3 (CDR3β) loop of the Vβ domain that is crucial for peptide recognition (*Ding et al., 1998*; *Bourcier et al., 2001*; *Rudolph et al., 2006*). In crystal structures, both TCRs bind in a diagonal orientation to the Tax peptide-bound major histocompatibility complex (pMHC), such that both Vα and Vβ contact the MHC α1 and α2 helices (*Garboczi et al., 1996a*). The similar diagonal binding modes are achieved by interactions involving different CDR residues of A6 and B7 contacting largely the same sets of pMHC residues (*Ding et al., 1998*). Only 1 out of 17 residues contacting pMHC in B7 is also found in the A6 pMHC interaction (*Ding et al., 1998*). T-cell response assays demonstrated that single-residue mutations to the Tax peptide elicit different responses in the two TCRs (*Ding et al., 1998*; *Hausmann et al., 1999*). Interfacial interactions may play a role in this TCR-specific response, as residue charges at the surface of the A6 and B7 variable domains show different electrostatic profiles, where the pocket for the Tax peptide Y5 residue is positively charged in A6 but negatively charged in B7 (*Ding et al., 1998*). Overall, the B7 Vα surface has fewer charged residues exposed than A6 Vα. While A6 and B7 recognize Tax-MHC with similar affinities and kinetics, it has been suggested that they achieve binding via different thermodynamic pathways (*Davis-Harrison et al., 2005*).

However, since αβTCR is a mechanosensor (*Reinherz et al., 2023*), the TCR-pMHC bond lifetime under physiological piconewton (pN) level load should be more functionally relevant than the

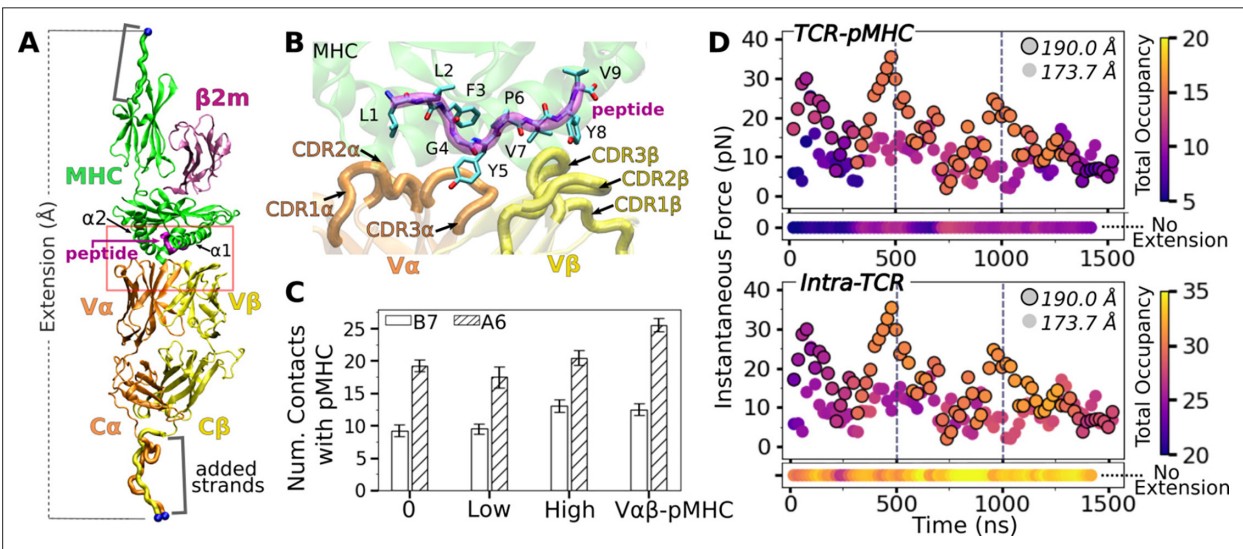

**Figure 1.** B7 T-cell receptor (TCR)-peptide-bound major histocompatibility complex (pMHC) interface. (**A**) Overview of the base complex used in simulations. The four subdomains of TCRαβ are Vα, Vβ, Cα, and Cβ. Load was applied by holding the $C_\alpha$ atoms of terminal residues (blue spheres at the ends of 'added strands') at a given distance from each other. β2m: β2 microglobulin. (**B**) Magnified view of red box in panel A showing labeled CDR loops and side chains of peptide residues in stick representation. (**C**) Number of contacts with greater than 50% average occupancy and 80% maximum instantaneous occupancy from 500 to 1000 ns. Bars: std. Criteria for counting contacts and values for A6 are from *Chang-Gonzalez et al., 2024*. (**D**) Total contact occupancy measured in 40 ns overlapping intervals. TCR-pMHC (top; intermolecular) and intra-TCR (bottom; intramolecular) contacts are shown separately. Intra-TCR contacts exclude Cα-Cβ contacts (Methods). Circles with outline: B7[high]; without outline: B7[low]. Horizontal bar below each panel: B7[0].

The online version of this article includes the following figure supplement(s) for figure 1:

**Figure supplement 1.** Model of T-cell receptor (TCR)αβ with leucine zipper domain fused to the C-terminal added strands.

**Figure supplement 2.** Standard deviation vs. average of the applied force.

**Table 1.** Simulation systems constructed based on PDB 1BD2 (*Ding et al., 1998*).
Extension is the distance between the harmonic potentials on the terminal restrained atoms (blue spheres in *Figure 1A*), which was selected for B7$^{low}$ and B7$^{high}$ to yield average low and high loads among simulations scanning different extensions (see Methods). Average load is calculated between 500 and 1000 ns. The standard deviation (std) in load as measured in 40 ns intervals between 500 and 1000 ns is shown in parentheses. Their values are consistent with those for A6 (*Figure 1—figure supplement 2*). The extension and force (average±std) for A6 corresponding to B7$^{low}$ and B7$^{high}$ are 182.6 Å and 13.2±5.65 pN and 187.7 Å and 18.2±9.17 pN, respectively (*Chang-Gonzalez et al., 2024*), which indicates that B7 is more compliant compared to A6. For B7$^0$, B7$^{low}$, and B7$^{high}$, simulations were further extended for additional time-dependent stability analysis. However, averaging was done for the 500–1000 ns interval for consistency.

| Label | Structure | Time (ns) | Extension (Å) | Load (pN) |
|---|---|---|---|---|
| Vαβ | Vα-Vβ only (no pMHC) | 1000 | – | – |
| Tαβ | TCRαβ only (no pMHC) | 1000 | – | – |
| Vαβ-pMHC | Vαβ with pMHC (no C-module) | 1000 | – | – |
| B7$^0$ | TCRαβ-pMHC | 1450 | – | – |
| B7$^{low}$ | TCRαβ-pMHC | 1550 | 173.7 | 9.01 (3.96) |
| B7$^{high}$ | TCRαβ-pMHC | 1550 | 190.0 | 14.5 (7.20) |

equilibrium binding pathway. In this regard, we have previously used all-atom molecular dynamics (MD) simulations to show that contacts with pMHC of A6 (*Chang-Gonzalez et al., 2024*) and JM22 (*Hwang et al., 2020*) TCRs are stabilized when an adequate 15–20 pN force is applied. The force-induced stabilization occurs as the asymmetric domain motion of the TCR chassis leads to weakening of the interface with pMHC either in the absence of an adequate level of force or if the sequence of the bound peptide is incompatible with maintaining contacts in the loaded state. The goal of this study is to determine whether the load-dependent control of the binding with pMHC is also present in B7 TCR, and identify any differences with A6 that may impact the response of the T-cell while responding to the same pMHC.

We find that the mechanism of dynamic allostery is largely conserved in B7, yet the loaded state does not stabilize contacts with pMHC as robustly as in the A6. Thus, while both A6 and B7 possess comparable equilibrium binding affinity for pMHC, A6 appears to exhibit a stronger catch bond behavior under load. Given the nuanced and dynamic nature of TCRs against the same pMHC, including the effect of ligand abundance (*Akitsu et al., 2024*), difference in mechanical response between A6 and B7 suggests T-cell clones bearing those TCRs may operate differently in vivo. Our study also underscores the importance of dynamics when comparing between TCRs that have similar crystal structures and equilibrium binding affinities.

## Results

Our simulation systems include the B7 TCR bound to Tax pMHC (*Figure 1A and B*) with no, low, and high extensions to apply different loads, and an isolated B7 TCR (*Table 1*). We also used systems without the C-module (Vαβ and Vαβ-pMHC) to study the role of the C-module. To apply a load, we held the terminal C$_\alpha$ atoms of the added strands at different extensions (*Figure 1A*). This reflects the constant spacing between a T-cell and an APC maintained by adhesion molecules such as CD2 and CD58 (*Reinherz et al., 2023*). In in vitro single-molecule experiments, pulling to a fixed separation and holding is also commonly done. On the other hand, simulation for B7$^0$ was performed by lightly holding the MHC α3 domain (*Figure 1A*) in a construct without added strands (Methods). It thus does not have any extension nor applied load.

We analyzed TCR-pMHC intermolecular and intra-TCR (intramolecular) interactions and domain motion to test whether the TCR-pMHC interface is stabilized by load and to find the underlying allostery mechanism that involves the motion of the TCR chassis. We analyze dynamics during the full trajectory and average behavior during 500–1000 ns.

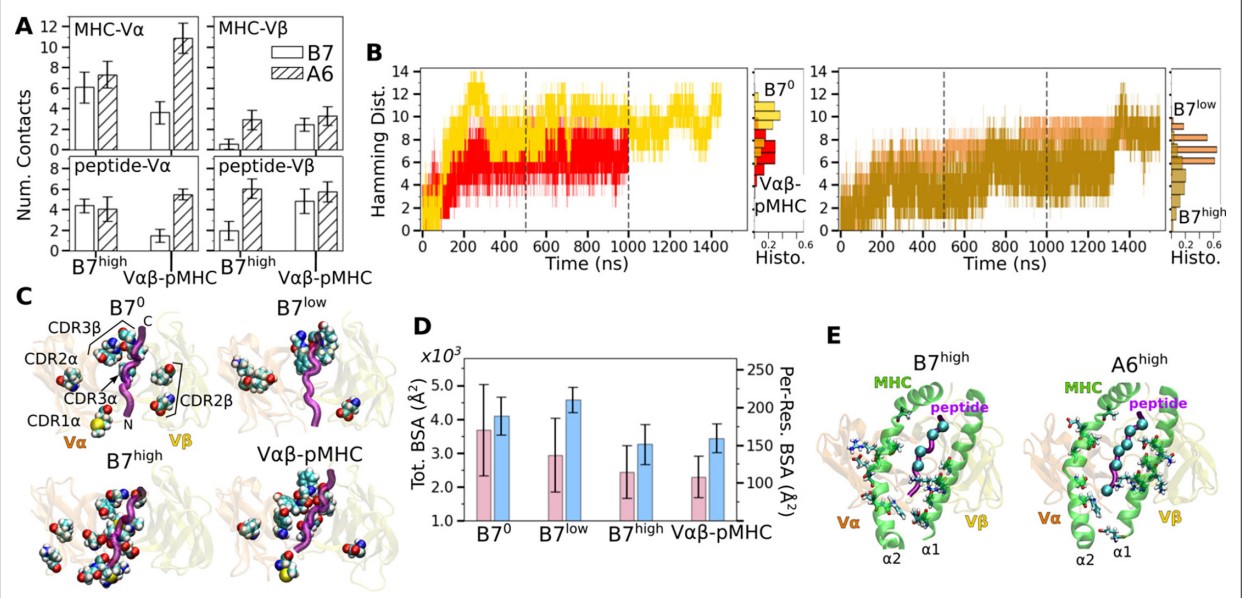

**Figure 2.** Load-dependent behavior of the B7 T-cell receptor (TCR)-peptide-bound major histocompatibility complex (pMHC) interface. (**A**) Number of MHC-Vα, MHC-Vβ, peptide-Vα, and peptide-Vβ contacts in B7 and A6 TCRs between 500 and 1000 ns. Data for A6 are from ***Chang-Gonzalez et al., 2024***. (**B**) Hamming distance $\mathcal{H}$. Histograms were calculated using data between 500 and 1000 ns (marked by vertical dashed lines). (**C**) Location of V-module residues forming contacts with pMHC with greater than 50% average occupancy. The frame at 1000 ns is used for visualization. The backbone of the Tax peptide is shown as a purple tube. CDRs are labeled in the first panel. (**D**) Total (pink) and per-residue (blue) buried surface area (BSA) for interfacial residues between 500 and 1000 ns. (**E**) pMHC residues forming contacts with the V-module with average occupancy greater than 50% in the high-load case. Left: B7[high], right: A6[high]. MHC residues are shown as sticks and $C_\alpha$ atoms of the peptide residues are shown as spheres. Viewing direction is the same as in panel C.

The online version of this article includes the following figure supplement(s) for figure 2:

**Figure supplement 1.** Additional characterization of the T-cell receptor (TCR)-peptide-bound major histocompatibility complex (pMHC) interface.

## Stabilization of the TCRαβ-pMHC interface with load

Compared to the no- (B7[0]) and low-load (B7[low]) cases, the number of high-occupancy contacts were more numerous for the high-load case (B7[high]), indicative of a catch bond behavior. Absence of the C-module (Vαβ-pMHC) also promoted more contacts with pMHC, suggesting the allosteric role of the C-module for binding with pMHC (***Figure 1C***). These results agree well with the behaviors seen in A6 and JM22 TCRs in our previous studies (***Chang-Gonzalez et al., 2024***; ***Hwang et al., 2020***). However, the number of pMHC contacts with B7 was reduced compared to A6 (***Figure 1C***). This is despite their comparable number of contacts with the pMHC in crystal structures and comparable equilibrium binding affinity in solution (***Ding et al., 1998***; ***Davis-Harrison et al., 2005***). In fact, we had to modify our simulation protocol to avoid premature breakage of contacts between B7 and pMHC when preparing the system for production run under load (see Methods).

Time-dependent behavior of the TCR-pMHC interface further supports the load-mediated enhancement of binding. For this, we calculated instantaneous force and contact occupancy in 40 ns overlapping intervals (***Figure 1D***). B7[high] had overall more inter- and intramolecular contacts (more orange-yellow circles) than B7[low] (more purple), suggesting that the increased load stiffens the TCR-pMHC complex. B7[0] had fewer TCR-pMHC contacts but more intra-TCR contacts (horizontal bars in ***Figure 1D***), indicating decoupling of the TCR from pMHC in the absence of load. However, after about 1300 ns, B7[high] started to lose the initial high-occupancy contacts (***Figure 1D***). This is likely because the fixed extension (end-to-end distance) imposed in simulation can adversely impact the interfacial stability of the B7 TCR-pMHC complex that has a higher compliance than A6, as discussed further below. For analysis of average properties, we thus used the 500–1000 ns interval, to be consistent with other systems.

Comparing average counts of high-occupancy pMHC contacts for both A6 and B7 TCRs indicates that interfacial contacts were dominated by MHC-Vα (***Figure 2A***). Also, Vα formed more contacts with

MHC than Vβ, while Vβ formed more contacts with the peptide than it does with MHC (*Figure 2A*). Similar to A6 (*Chang-Gonzalez et al., 2024*), occupancy heat maps for individual contact residue pairs show reduced or fragmented contacts for $B7^0$ and $B7^{low}$ (*Figure 2—figure supplement 1A and B*, more red compared to blue) while $B7^{high}$ and Vαβ-pMHC exhibit more persistent contact profiles (*Figure 2—figure supplement 1C and D*, more blue compared to red).

Temporal progression of the number of contacts was measured via the Hamming distance $\mathcal{H}$, the number of the initial high-occupancy contacts lost over time (*Figure 2B*; see Methods). For $B7^0$, $\mathcal{H}$ rapidly increased and by 200 ns, most of the initial high-occupancy contacts were lost. While the increase in $\mathcal{H}$ for Vαβ-pMHC was comparable to that of $B7^{low}$ (*Figure 2B*), the contact occupancy heat maps reveal that Vαβ-pMHC maintained contacts after a brief initial adjustment (arrow in *Figure 2— figure supplement 1D*) while contacts were lost in $B7^{low}$ (red in *Figure 2—figure supplement 1B*). On the other hand, $B7^{high}$ maintained $\mathcal{H}$ comparable to Vαβ-pMHC until about 1300 ns, after which it approached that of $B7^0$. This occurred as some of the high-occupancy contacts broke (*Figure 2—figure supplement 1C*). Importantly, the breakage is not due to a high force, but instead it happened when instantaneous force was low. Low-force states are reached at around 750 and 1300 ns (*Figure 1D*), followed by stepwise increase in $\mathcal{H}$ (*Figure 2B*). The difference in extension between $B7^{low}$ and $B7^{high}$ is 16.3 Å whereas it is 5.1 Å for A6 (*Table 1*), for similar ~5 pN difference between low- and high-load cases. Being more compliant, the interface of B7 can reach a low-force state more easily, hence it is more prone to destabilization.

Location of V-module residues forming contacts with pMHC with greater than 50% average occupancy were concentrated along the peptide for $B7^{high}$ and Vαβ-pMHC, but scattered in $B7^0$ and $B7^{low}$ (*Figure 2C*). This trend was also observed in A6 and JM22. Such concentration of high-occupancy contacts may protect TCR-pMHC interactions from breakage by water. In A6 and JM22 TCRs, the greater number of contacts with pMHC in the high-load cases and Vαβ-pMHC correlated with larger buried surface area (BSA) of the residues forming contacts (*Chang-Gonzalez et al., 2024*; *Hwang et al., 2020*). B7 did not follow this trend, as $B7^{high}$ and Vαβ-pMHC had reduced total and per-residue BSA than $B7^0$ and $B7^{low}$ (*Figure 2D*). This is likely because the fewer high-occupancy contacts in B7 (*Figure 1C*) tend to be more exposed, making the relationship between the BSA and load less direct. Consistent with this, the total BSA of B7 was 67.4% ($B7^0$) to 44.2% ($B7^{high}$) of the corresponding values of A6.

The residues of pMHC forming greater than 50% average occupancy under high load differed between A6 and B7 (*Figure 2E*), as did the location of V-module residues forming respective contacts with the pMHC residues (*Figure 2C*), further highlighting their divergence in the interfacial footprint under load. Despite this, the root mean square fluctuation (RMSF) of $C_\alpha$ atoms of the Tax peptide measured after 500 ns was reduced in $B7^{high}$ and Vαβ-pMHC compared to $B7^0$ and $B7^{low}$

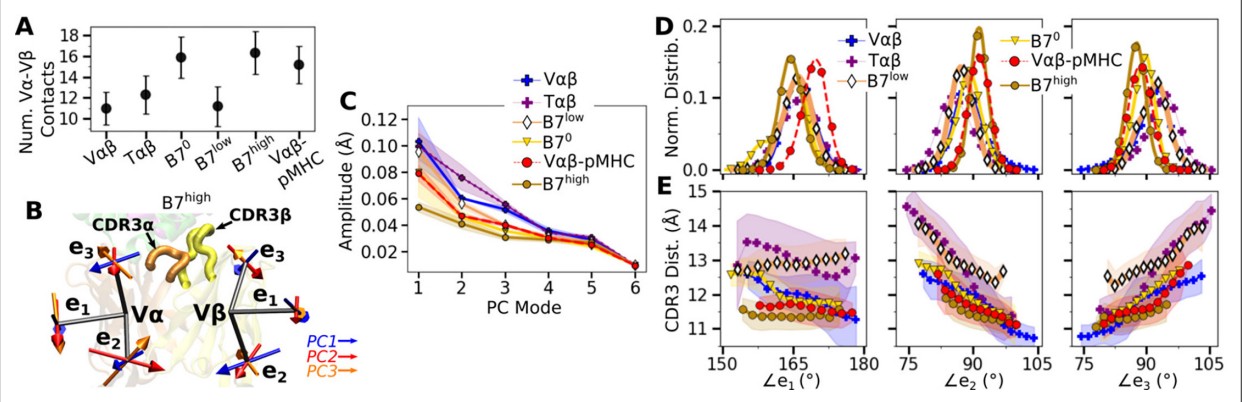

**Figure 3.** Vα-Vβ motion of B7. (**A**) Number of Vα-Vβ contacts with greater than 50% average occupancy and 80% maximum instantaneous occupancy between 500 and 1000 ns. Bars: std. (**B**) V-module triads $\{\mathbf{e}_1, \mathbf{e}_2, \mathbf{e}_3\}$. Arrows denote directions of the first three PC modes for $B7^{high}$ as an example. CDR3s are labeled. (**C**) Amplitudes for the first six PCs. Principal component analysis (PCA) was performed between 500 and 1000 ns. Transparent bands: std for PCA performed in three overlapping intervals (500–800 ns, 600–900 ns, and 700–1000 ns). (**D**) Histograms of the V-module triad angles. (**E**) CDR3 distance vs. triad angles. Transparent bands: std. Both D and E were determined from data between 500 and 1000 ns.

(*Figure 2—figure supplement 1E*), as observed for A6 (*Chang-Gonzalez et al., 2024*), which supports load-induced stabilization of the complex.

We calculated the distance between the V-module and pMHC as another measure of the interfacial stability (*Figure 2—figure supplement 1F*; Methods). The distance was stably maintained in Vαβ-pMHC and B7$^{high}$ (before 1300 ns) whereas it fluctuated more in B7$^0$ and B7$^{low}$. Of note, the former two systems maintained the distance greater than that in the crystal structure by 0.3–0.9 Å. Thus, a slight separation engendered by force or in the absence of constraint imposed by the C-module provides room for adjusting residues to form more stable contacts. The more stable maintenance of the distance between V-module and pMHC in Vαβ-pMHC and B7$^{high}$ is consistent with other measures of their stability explained above.

## CDR3 positions are controlled by load-dependent Vα-Vβ motion

The greater number of Vα-Vβ contacts in B7$^0$ (*Figure 3A*) is consistent with the increase in total intra-TCR contact occupancy (horizontal bar in *Figure 1D*, bottom panel). Without load this does not translate to a stronger TCR-pMHC interface explained above. B7 in general had fewer Vα-Vβ contacts (11.0–16.3) than A6 (15.9–23.1) (*Chang-Gonzalez et al., 2024*). The ~70% reduction in Vα-Vβ contacts for B7 is comparable to the ~50% reduction in contacts with pMHC between the two TCRs (*Figure 1C*).

Vα-Vβ motion was measured via triads (orientational markers) assigned to respective domains and by performing principal component analysis (PCA) (*Figure 3B*; Methods). PC amplitude was the lowest for B7$^{high}$ and Vαβ-pMHC (*Figure 3C*), which is consistent with the greater number of Vα-Vβ contacts. Regarding the direction of motion, the mutually orthogonal PC directions can be difficult to interpret (arrows in *Figure 3B*). We instead measured angles between the matching arms of the two triads named $\angle e_i$ ($i = 1, 2, 3$), to examine the Vα-Vβ motion in structurally interpretable directions

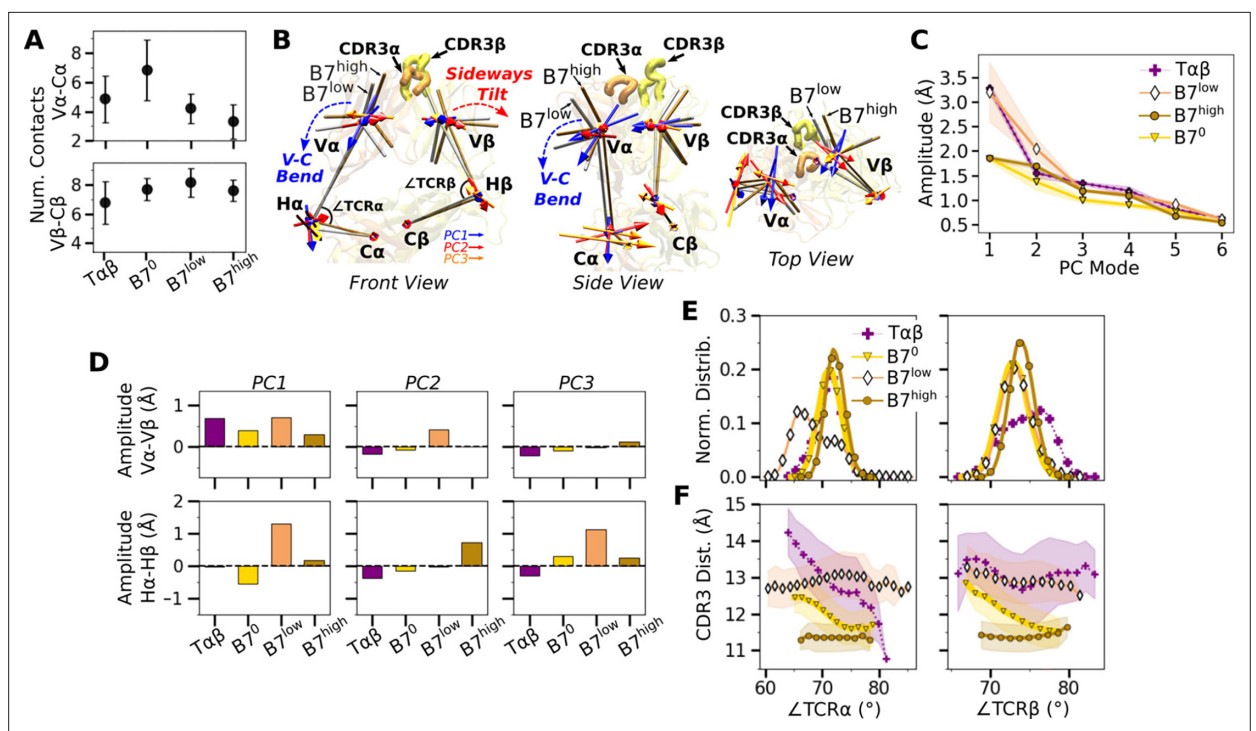

**Figure 4.** V-C motion of B7. (**A**) Number of V-C contacts per chain measured with the same criteria as *Figure 3A*. (**B**) Average bead-on-chain (BOC) for B7$^{low}$ and B7$^{high}$. The V-module of B7$^{high}$ is less bent compared to B7$^{low}$. The arrows for the first three V-C PC modes are shown, where PC1 corresponds to the V-C bending motion. (**C**) V-C PC amplitudes for the first six PC modes. Transparent bands: std measured in the same way as in *Figure 3C*. (**D**) Differences in amplitudes for the first three PCs between matching V- and H-beads of α and β chains. (**E**) Histograms of hinge angles defined in panel B. (**F**) CDR3 distance vs. hinge angles. Transparent bands: std. All panels are generated with data between 500 and 1000 ns.

The online version of this article includes the following figure supplement(s) for figure 4:

**Figure supplement 1.** Additional characterization of the V-C motion.

(*Figure 3D*; *Chang-Gonzalez et al., 2024*). For example, $\angle e_1$ is the angle between each $\mathbf{e}_1$ arm from Vα and Vβ, which describes a 'flapping' or 'twisting' motion of the two domains. Since $\mathbf{e}_2$ and $\mathbf{e}_3$ lie approximately parallel to the Vα-Vβ interface, they vary reciprocally, corresponding to a 'scissoring' motion (*Hwang et al., 2020*).

Measuring the distance between CDR3α and CDR3β ('CDR3 distance') revealed that this distance is the shortest for B7$^{high}$ followed by Vαβ-pMHC. Comparing CDR3 distance vs. triad angles (*Figure 3E*) shows that CDR3 distance varied in opposite directions with $\angle e_2$ and $\angle e_3$, which reflects their reciprocal relation (opposite slopes in *Figure 3E*). In comparison, the CDR3 distance of the B7 crystal structure is 12.0 Å, which is larger than those of B7$^{high}$ and Vαβ-pMHC (*Figure 3E*). The slight separation between the V-module and pMHC (*Figure 2—figure supplement 1F*) in Vαβ-pMHC and B7$^{high}$ allows CDR3 loops to come closer together compared to the crystal structure, akin to pinching the central protrusion of the peptide.

## Asymmetric V-C bending in the B7 TCR is suppressed with applied load

Next, we considered the motion between the V- and C-modules ('V-C motion'). The number of high-occupancy contacts for the Cα-Cβ interface (26.2–27.5) was considerably greater than those for Vα-Vβ (11.2–16.3), indicating that the C-module acts as a single base for the V-C motion, as noted for A6 and JM22 TCRs (*Hwang et al., 2020*; *Chang-Gonzalez et al., 2024*). Continuing this general feature, there were fewer high-occupancy contacts for the Vα-Cα interface compared to the Vβ-Cβ interface (*Figure 4A*).

The V-C motion was analyzed by using the bead-on-chain (BOC) model that tracks individual domains and the hinge between them (*Figure 4B*). PC motion directions were compared by calculating the dot products between the corresponding PC vectors (*Figure 4—figure supplement 1A*). PC1 corresponding to the V-C bending in B7$^{high}$ (*Figure 4B*) was similar in other systems, whereas B7$^{low}$ differed the most (darker color for B7$^{low}$ in *Figure 4—figure supplement 1A*, PC1). Amplitudes of PC1 show a clear distinction where the unliganded Tαβ and B7$^{low}$ were more mobile than B7$^0$ and B7$^{high}$ (*Figure 4C*). Comparing PC amplitudes of the elements of the BOC between α and β chains revealed that Vα moves more relative to the C-module than Vβ (*Figure 4D*), similar to A6 (*Chang-Gonzalez et al., 2024*). Amplitude of the hinge motion in the two chains varied, where Hα had greater amplitude in PC1 for B7$^{low}$ compared to B7$^{high}$ (*Figure 4D*, PC1 in bottom row). This suggests a more pronounced asymmetric motion in B7$^{low}$. The no-load B7$^0$ Hβ amplitude was larger compared to Hα (*Figure 4D*, bottom, negative value for PC1 of B7$^0$). For B7$^0$, the small PC1 amplitude of the overall V-C motion without load (*Figure 4C*) does not suppress the motional asymmetry between α and β chains, while in B7$^{high}$, the chassis becomes less mobile under load.

The V-C angles (*Figure 4B*, ∠TCRα and ∠TCRβ) reveal the motional asymmetry in addition to PCA. As in the case of low-load A6 (*Chang-Gonzalez et al., 2024*), ∠TCRα of B7$^{low}$ shows a wide bimodal distribution (*Figure 4E*). ∠TCRα decreased at around 700 ns as B7$^{low}$ bent more (*Figure 4—figure supplement 1B*). This would put the V-module in an unfavorable orientation to bind pMHC (*Hwang et al., 2020*). The dependence of CDR3 distance on V-C angles was most steady in B7$^{high}$ (*Figure 4F*). This supports the allosteric mechanism by which the asymmetric V-C motion of the whole TCR controls the Vα-Vβ motion and the V-C orientation that in turn affects the stability of the TCR-pMHC interface.

## Discussion

As a protein-protein complex, TCR-pMHC forms a weak interface. A typical heterodimeric protein-protein interface with BSA comparable to that of TCR-pMHC (~1700 Å$^2$) has sub-μM binding affinity (*Chen et al., 2013*) while the affinity of the TCR-pMHC complex ranges between μM to hundreds of μM (*Rudolph and Wilson, 2002*; *Rudolph et al., 2006*). Given the low equilibrium affinity, recent findings highlight the importance of mechanosensing, where force generated during immune surveillance of αβ T-cells is utilized to discriminate cognate vs. non-cognate pMHCs (reviewed in *Zhu et al., 2019*; *Liu et al., 2021*; *Reinherz et al., 2023*). Our earlier simulation studies of the JM22 and A6 TCRs showed that the dynamical motion of the TCR chassis is responsible for the force-driven stabilization of the contacts with pMHC, i.e., catch bond formation (*Hwang et al., 2020*; *Chang-Gonzalez et al., 2024*). Examining additional TCRs via all-atom MD simulation can inform how the mechanism is adopted. By finding similarities and differences between A6 and B7 TCRs, which recognize the

same Tax pMHC, we can elucidate how mechanical force is utilized by each to impact the differential function of T-cells.

Catch bond mechanisms generally differ among proteins (*Thomas et al., 2008*). For example, FimH and vinculin have relatively well-defined weak- and strong-binding states where there are corresponding crystal structures (*Bakolitsa et al., 1999*; *Le Trong et al., 2010*; *Sauer et al., 2016*; *Mei et al., 2020*). Availability of the end-state structures enable using simulation approaches such as enhanced sampling of individual states and studying the transition between the two states (*Languin-Cattoën et al., 2023*; *Ccoa et al., 2024*). In contrast, TCRs do not have any structurally well-defined weak- or strong-binding states, which requires a different approach. As demonstrated in our present work as well as in our previous papers (*Hwang et al., 2020*; *Chang-Gonzalez et al., 2024*), our microsecond-long simulations of the complex under realistic pN-level loads and a combination of analysis methods are effective for elucidating the catch bond mechanism of TCRs, which shows that the load-dependent stabilization of the TCR-pMHC complex differs from simple relaxation of the structure in the initial crystallographic confinement.

Recently, *Choi et al., 2023*, proposed a mathematical model of TCR catch bond formation. Their assumptions are based mainly on the steered MD simulation (*Wu et al., 2019*) where partial unfolding of MHC and tilting of the TCR-pMHC interface were considered as required. Our mechanism does not conflict with their assumptions since the complex in the fully folded state should first bear load in a ligand-dependent manner in order to allow any subsequent larger-scale changes.

We found that the dynamic allostery of the TCR chassis observed in A6 (*Chang-Gonzalez et al., 2024*) and JM22 (*Hwang et al., 2020*) is largely conserved in B7. Cα and Cβ domains form extensive contacts to construct a base, and the Vβ-Cβ contacts are more extensive than Vα-Cα contacts. This leads to an asymmetric V-C motion where Vα is more mobile relative to the C-module than Vβ. In turn, the motional asymmetry affects the relative positioning of CDR loops, as measured by the CDR3 distance as well as the orientation of the TCR-pMHC interface relative to the loading direction. Unless an adequate load is applied to the complex to suppress the motion, destabilization of the interface occurs. As noted in A6 and JM22, the C domain renders a disadvantage to binding except under force. This is indicated by the greater number of interfacial contacts and decreased Vα-Vβ motion of Vαβ-pMHC (no C-module) simulations. The increased interfacial stability of Vαβ-pMHC is also consistent with our discovery that the C-module likely undergoes a partial unfolding to an extended state, where the bond lifetime increases (*Das et al., 2015*; *Akitsu et al., 2024*). Therefore, while the variable domain dictates TCR fit with pMHC, a logical evolutionary pressure would be for the constant domain to maximize discriminatory power by adding instability to the TCR chassis. Furthermore, considering single-chain versions of an antibody lacking the C-module (scFv) are in widespread use (*Ahmad et al., 2012*) including CAR T cells (*Reinherz et al., 2023*), a better understanding of a TCR lacking the C-module may help with developing a novel TCR-based immunotherapy. Given the conservation of the asymmetric motion, we also predict the Cβ FG-loop deletion mutant of B7 TCR to possess a diminished catch-bond response, as our previous experiment (*Das et al., 2015*) and simulation *Hwang et al., 2020*; *Chang-Gonzalez et al., 2024* demonstrated for other TCRs.

While dynamic allostery is overall conserved, A6 and B7 differ in the behavior of the TCR-pMHC interface under load. The crystal structures of the two complexes have a comparable number of TCR-pMHC contacts, with a total of 33 for A6 and 29 for B7 in terms of residue pairs, while the number of distinct atom pairs forming contacts are 46 for A6 (*Garboczi et al., 1996a*) and 63 for B7 (*Ding et al., 1998*). During MD simulations, many of these contacts become transient, resulting in fewer high-occupancy contacts. In the thermally fluctuating state, there is a fourfold increase in the difference in number of contacts between A6 and B7 over the difference for static crystal structures (*Figure 1C*). Previously, contacts found in crystal structures have been used to explain differences in responses to point mutations to the Tax peptide. For example, the sensitivity of B7 to the mutation of Y5 of Tax was explained based on the Y5-D30α hydrogen bond and stacking of Y5 with Y101β of B7 (Y101β in the present study was numbered Y104β in *Ding et al., 1998*, and *Hausmann et al., 1999*). In comparison, the more amenable pocket for Y5 on A6 was suggested to be responsible for the greater tolerance of A6 under mutations of Y5. In MD simulations of B7, the Y5-D30α hydrogen bond is formed only in B7$^{high}$, after about 300 ns (*Figure 2—figure supplement 1C*), and the Y5-Y101β nonpolar contact breaks during simulation of B7$^0$ and is not present in B7$^{low}$ (*Figure 2—figure supplement 1A and B*) while it persists with 62% occupancy in B7$^{high}$ and 52% occupancy in Vαβ-pMHC (*Figure 2C and D*;

these occupancies are for the whole simulation period). Changes to T-cell specificity by point mutations on Y5 are thus unlikely to arise solely from the size or the polarity of its binding pocket as a static structure. Instead, point mutations affect the organization and dynamics of the surrounding contacts in addition to fit (*Chang-Gonzalez et al., 2024*). In a related vein, intermolecular motional network has been suggested to be a controlling factor for allostery in a peptide-SH3 domain complex (*Gomez et al., 2024*), which aligns with the nonlocal and dynamic role of contacts at the TCR-pMHC interface. The inter-domain motion of TCR enables long-range allosteric discrimination of pMHCs, integrating the motion of the two entities such that they influence each other.

The 14.5 pN force on B7$^{high}$ was applied under 190.0 Å extension (*Table 1*), whereas the corresponding high-load simulation for A6 was with 18.2 pN at a slightly lower 187.7 Å extension (*Chang-Gonzalez et al., 2024*). The average total intra-TCR contact occupancy for B7$^{high}$ between 500 and 1000 ns is 30.4±0.49 contacts (*Figure 1C*), compared to 38.7±0.87 for A6 under high load (after 500 ns), which further supports a higher compliance of the former. With its higher compliance, the B7 TCR-pMHC complex needs to be under a greater extension than A6 to apply comparable levels of force, and it would be more difficult to achieve load-induced stabilization of the TCR-pMHC interface. This leads us to predict that B7 will exhibit a weaker catch bond where the peak of the TCR-pMHC bond lifetime is at a lower force. And the destabilization of B7$^{high}$ at 1300 ns (*Figures 1D and 2B*) may have been a result of our constant-extension approach to apply load, which allows reaching low-force states by the compliant B7. Although this approach was motivated by the constant spacing between a T-cell and an APC (*Reinherz et al., 2023*), the actin cytoskeleton also adjusts the load applied to the TCR-pMHC complex (*Feng et al., 2017*; *Hu et al., 2024*). How fluctuation in the applied load affects the TCR-pMHC dynamics is a subject of a future study.

As a related example to A6 and B7 TCRs, PA59 and PA25 TCRs share the same *TRBV* and *TRBJ* genes, and they share 11-aa CDR3β loops that differ only at a single position (Trp vs. Leu) (*Akitsu et al., 2024*). Recognizing the same pMHC (PA$_{224-233}$/D$^b$), their maximum bond lifetimes and the peak catch bond forces differ considerably, 75 s at 21 pN for PA59, and 13 s at 15 pN for PA25 (*Akitsu et al., 2024*). These differences may facilitate actions of the corresponding T-cells in tissues that present diverse mechanical environments provided by cell movement, adhesion molecules, tight junction, and noncompliant inflammation.

Likewise, T-cells bearing A6 and B7 may also perform differently in vivo depending on tissue localization (*Akitsu et al., 2024*) even though they behave similarly in vitro in terms of cytotoxicity and secretion of select cytokines (γ-IFN, MIP-1α, and TNFα) (*Ding et al., 1998*; *Hausmann et al., 1999*). We also predict the different catch bond profiles of A6 and B7 will render the two types of T cells to respond differently when the pMHC copy number on the APC is limited (*Akitsu et al., 2024*). Differences in the catch-bond profiles may also affect the energy input for downstream signaling events that involve reversible conformational transitions in TCRαβ within a certain range of forces and extends the TCR-pMHC bond lifetime by more than an order of magnitude (*Akitsu et al., 2024*). Elucidating details of those differences is beyond the scope of the present work.

Disparate biological outcomes between structurally similar TCRs recognizing the same pMHC are mechanically possible since slight changes in interfacial contacts can result in altered distribution of loads across the TCR chassis so that changes in its dynamics affect interaction with CD3 signaling subunits (*Reinherz et al., 2023*). Structural details of the dynamic amplification and propagation of the recognition signal warrant further investigation.

## Methods
### Structure preparation

B7 TCRαβ-pMHC was built from PDB 1BD2 (*Ding et al., 1998*) using CHARMM (*Brooks et al., 2009*; *Hwang et al., 2024*). Non-numeral residue IDs in the PDB were renumbered to follow sequential numbering used in the present study. We used MODELLER (*Sali and Blundell, 1993*) to generate coordinates for missing loops in the Cα domain (S133-K136 and S170-D172 in the PDB numbering scheme) followed by a brief energy minimization. We visually verified MODELLER results, comparing generated loops to those of the related A6 TCR (*Garboczi et al., 1996a*; *Ding et al., 1998*). The constant domain of TCRα (Cα) was also missing coordinates for F204-S210 (F198-S204 after renumbering), which were added with the TCRα linker as detailed below. Disulfide bonds were assigned

between cysteine residues as defined in the PDB file. Crystal waters within 2.8 Å from the protein were kept for the truncated structures, and all waters were kept for the full structure.

Histidine protonation state was determined to promote hydrogen bond formation with neighboring residues. The histidine $N^\delta$ atom was protonated as follows: MHC residues 3, 93, 114, 145, 151, 188, 260; β2m residues 13, 51; TCRα all histidine residues; and TCRβ residues 29, 47, 154. For the remaining histidine residues, the $N^\epsilon$ atom was protonated.

As done in *Hwang et al., 2020*, and *Chang-Gonzalez et al., 2024*, we extended the MHC and TCRαβ termini as handles for applying positional restraints. For MHC, we used UniProt P01892 to add $^{276}$LSSQPTIPI$^{284}$. For TCRα we used GenBank AAA60627.1 to add $^{205}$CDVKLVEKSFETDT$^{218}$. For TCRβ we used GenBank AAC08953.1 to add $^{245}$CGFTSESYQQGVLSA$^{259}$. We placed an interchain disulfide bond between αC205-βC245. Added strands in the initially straight conformations were relaxed to a state similar to that in *Figure 1A* by performing a series of brief energy minimization and MD simulation with the FACTS implicit solvent model (*Haberthür and Caflisch, 2008*).

Truncated structures were built based on the prepared B7 TCR-pMHC complex as:

- Vαβ: The last residues were αP110 and βV113.
- Tαβ: The last residues were αD206 and βG246 (no C-terminal strands).
- Vαβ-pMHC: includes Vαβ, pMHC, and β2m. The last residue of MHC was L276.
- B7$^0$: includes Tαβ, pMHC, and β2m. The last residue of MHC was L276.

## MD simulation protocol

Solvation, energy minimization, heating, and equilibration of the B7 complexes followed the protocol in *Chang-Gonzalez et al., 2024*, except for systems which include the pMHC, where we modified the preparation protocol prior to production runs as detailed below.

### Laddered extensions

Applying the same protocol as done for A6 to achieve the laddered extensions in B7 resulted in substantial breakage of the TCR-pMHC contacts within the first 50 ns in several production runs. To mitigate this, we introduced distance restraints to selected atom pairs forming contacts between the TCR and pMHC to prevent them from breaking during preparatory simulations. This ensured that the complex could structurally adapt as we modified the extension distance, yet all laddered extensions maintained a core set of initial TCR-pMHC contacts. Atom-pair distance restraints were removed in production runs.

Twelve atom pairs between TCR and pMHC were selected that were within 5 Å of each other in the equilibration restart file of the TCR-pMHC complex with added linkers. A 2 kcal/[mol·Å²] flat-bottom harmonic restraint potential was applied to keep the atom-pair distance within the value at the end of the equilibration run. Then, a 2 ns CPT simulation was carried out while also applying a 1 kcal/[mol·Å²] harmonic potential to the $C_\alpha$ atoms of the C-terminal MHC (I284), TCRα (T218), and TCRβ (A259) residues. Production run followed upon removing the atom-pair distance restraints. During the production run, the 1 kcal/[mol·Å²] harmonic potential to the TCR and MHC end-residues and a 10 Å flat-bottom harmonic distance restraint between αT218 and βA259 were applied, the latter to prevent the ends of the C-terminal added strands from separating (*Figure 1—figure supplement 1*). Throughout the production run, we intermittently measured average and rolling force in 40 ns intervals and found these to be around 10 pN, an ideal target for the low-load simulation. This simulation is the 173.7 Å B7$^{low}$ system reported in *Table 1*.

Using the structure at the end of the 2 ns simulation with the TCR-pMHC atom-pair distance restraints, we increased the extension by 8 Å by moving the center of the 1 kcal/[mol·Å²] harmonic potential on the end-residue $C_\alpha$ atoms by 4 Å at each end. While keeping the TCR-pMHC atom-pair distance restraints applied, we launched another 2 ns simulation under the increased extension. Atom-pair distance restraints were then removed and the production run was launched. The extension averaged after 500 ns of this simulation was 181.7 Å. Following the same way, we increased the extension by another 8 Å, which led to the 190.0 Å B7$^{high}$ system reported in *Table 1*. We also decreased the extension by 8 Å from the initial 173.7 Å extension, which led to a 165.7 Å extension.

Among the four extensions tested, the 181.7 Å extension was not selected primarily because the average force of the simulation from 500 to 850 ns (the total length of the simulation) was 9.26 pN,

only barely higher than the reported load for B7$^{low}$. For the 165.7 Å simulation, the average force from 500 to 900 ns (total length of the simulation) was 15.7 pN. We had observed this high force at low extension for A6 TCR (*Chang-Gonzalez et al., 2024*) and attribute this to folding of the flexible added strands leading to contacts between the stands and the TCR constant domains.

### Vαβ-pMHC and B7$^0$

We also applied a 2 kcal/[mol·Å$^2$] flat-bottom harmonic distance restraint during preparatory simulations of Vαβ-pMHC and B7$^0$. We attempted to use the same set of atom pairs as in the laddered extension simulations, but considerable interface breakage occurred, likely due to changes in interfacial contacts after equilibration in these systems. We thus selected different atom pairs for Vαβ-pMHC and B7$^0$. For consistency, we selected 12 atom pairs, the same number as in the laddered extensions, and distributed in the same way between the Tax peptide or MHC residues to Vα or Vβ residues. The distance restraint was applied to the atom pairs for a 2 ns CPT simulation, then released for production runs.

### Systems without load

The following additional restraints were used for systems without load.

- Vαβ, Tαβ: no positional restraints were applied.
- Vαβ-pMHC: we applied a weak 0.01 kcal/[mol·Å$^2$] harmonic positional restraint to the backbone C$_\alpha$ atoms of MHC α3 (P185-L276) to prevent large transverse rotation of the whole molecule in the orthorhombic box.
- B7$^0$: we applied a 0.2 kcal/[mol·Å$^2$] harmonic positional restraint to the backbone C$_\alpha$ atoms of MHC α3 with RMSF less than 0.5 Å calculated from the simulations of B7$^{low}$. These residues were: L201-Y209, T240-Q242, T259-H263.

### Production runs

Production runs were performed similar to *Chang-Gonzalez et al., 2024*. We used OpenMM (*Eastman et al., 2017*) with the CHARMM param36 all-atom force field (*MacKerell et al., 2004*) and the particle-mesh Ewald method to calculate long-range electrostatic interactions. We used an Ewald error tolerance of 10$^{-4}$ which is 1/5 of the default value in OpenMM and a 12 Å cutoff distance for nonbonded interactions. The complexes were simulated at 300 K with a 2 fs time step using the Nose-Hoover integrator in OpenMM. Production run lengths are given in *Table 1*.

## Trajectory analyses

Analysis methods are detailed in *Chang-Gonzalez et al., 2024*. Below we mainly explain B7-specific residue selections. We used the frames for 500–1000 ns of the production runs to calculate the average and standard deviation of the number of contacts, BSA, CDR3 distance, PCA, and triad and V-C angles. With a coordinate saving rate of 20 ps, this leaves at least 25,000 frames for analysis.

### V-module to pMHC distance

The distance from TCR V-module to pMHC (*Figure 2—figure supplement 1F*) was measured between the center of mass of the C$_\alpha$ atoms of the same residues used to build the V-module triads (described below) to the center of mass of five C$_\alpha$ atoms from each of the central four strands forming the β-sheet floor located above the α1 and α2 helices of MHC (20 MHC atoms in total; *Figure 1A*). These were R6-T10, I23-Y27, Q96-G100, and Y113-A117. RMSF of these residues after 500 ns was below 1.4 Å in all B7 systems, so the measured distance is minimally affected by the intra-domain conformational motion.

### CDR3 distance

Distance between CDR3α and CDR3β (*Figures 3E and 4F*) was measured using the midpoint between backbone C$_\alpha$ atoms of two residues at the base of each CDR3, which are E93 and K97 for CDR3α and S94 and E102 for CDR3β.

## V-module triads

We assigned triads (*Figure 3B*) based on the backbone $C_\alpha$ atoms of the stably folded β-sheet core of each variable domain (*Hwang et al., 2020*; *Chang-Gonzalez et al., 2024*). Selected residues for triad assignment of the B7 systems were as follows. For Vα, I19-Y24, F32-K37, H71-I76, and Y87-M92. For Vβ, T20-Q25, M32-Q37, D73-L78, and Y89-S94. Prior to triad assignment we aligned all complexes to the first frame of B7$^{low}$ using the selected residues to monitor the relative motion between the two triads without global translation nor rotation.

## V-C BOC

We assigned BOCs (*Figure 4B*) as detailed in *Hwang et al., 2020*; *Chang-Gonzalez et al., 2024*. To place beads for the C-module, we used the following residues. For Cα, A118-R123, V132-D137, Y153-T158, and S171-S176. For Cβ, L143-T148, L157-N162, L190-R195, and F208-Q213. For the hinges, we used: αN114 for Hα, and βD116 and βL117 for Hβ. We aligned all complexes to the backbone $C_\alpha$ atoms of the selected residues of B7$^{low}$ then built BOCs to monitor the motion of the V-module relative to the C-module.

## Acknowledgements

This work was funded by US National Institutes of Health Grants P01AI143565 and R01AI136301. Simulations were performed by using computers at the Texas A&M High Performance Research Computing facility.

## Additional information

### Funding

| Funder | Grant reference number | Author |
|---|---|---|
| National Institute of Allergy and Infectious Diseases | P01AI143565 | Ana Cristina Chang-Gonzalez<br>Aoi Akitsu<br>Robert J Mallis<br>Matthew J Lang<br>Ellis L Reinherz<br>Wonmuk Hwang |
| National Institute of Allergy and Infectious Diseases | R01AI136301 | Ana Cristina Chang-Gonzalez<br>Aoi Akitsu<br>Robert J Mallis<br>Matthew J Lang<br>Ellis L Reinherz<br>Wonmuk Hwang |

The funders had no role in study design, data collection and interpretation, or the decision to submit the work for publication.

### Author contributions

Ana Cristina Chang-Gonzalez, Conceptualization, Data curation, Software, Formal analysis, Validation, Investigation, Visualization, Methodology, Writing - original draft, Writing – review and editing; Aoi Akitsu, Conceptualization, Writing – review and editing; Robert J Mallis, Conceptualization, Investigation, Writing – review and editing; Matthew J Lang, Ellis L Reinherz, Conceptualization, Funding acquisition, Investigation, Project administration, Writing – review and editing; Wonmuk Hwang, Conceptualization, Resources, Data curation, Software, Formal analysis, Supervision, Funding acquisition, Validation, Investigation, Visualization, Methodology, Writing - original draft, Project administration, Writing – review and editing

### Author ORCIDs

Ana Cristina Chang-Gonzalez https://orcid.org/0000-0002-1517-4172
Aoi Akitsu https://orcid.org/0000-0001-7672-9664

Robert J Mallis ⓘ https://orcid.org/0000-0002-2087-9468
Matthew J Lang ⓘ https://orcid.org/0000-0002-8198-144X
Ellis L Reinherz ⓘ https://orcid.org/0000-0003-1048-5526
Wonmuk Hwang ⓘ https://orcid.org/0000-0001-7514-3186

Reviewer #1 (Public review): https://doi.org/10.7554/eLife.104280.3.sa1
Reviewer #2 (Public review): https://doi.org/10.7554/eLife.104280.3.sa2
Reviewer #3 (Public review): https://doi.org/10.7554/eLife.104280.3.sa3
Author response https://doi.org/10.7554/eLife.104280.3.sa4

## Additional files

### Supplementary files
MDAR checklist

### Data availability
Sample simulation systems and scripts are available in GitHub (copy archived at *Hwang, 2025*).

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
