## [Editor Report · eLife Assessment]

This **useful** study reports detailed molecular dynamics (MD) simulations of T-cell receptors (TCRs) in complex with a peptide/MHC complex, for a better understanding of the mechanism of T-cell activation. The MD simulations provide **solid** evidence supporting that different TCRs can respond mechanically in different ways upon binding to the same pMHC complex. The analyses are systematic and provide testable predictions that can be evaluated by future mutagenesis and force microscopy studies.

---

## [Referee Report · Reviewer #1 (Public review)]

Summary:

This paper describes molecular dynamics simulations (MDS) of the dynamics of two T-cell receptors (TCRs) bound to the same major histocompatibility complex molecule loaded with the same peptide (pMHC). The two TCRs (A6 and B7) bind to the pMHC with similar affinity and kinetics, but employ different residue contacts. The main purpose of the study is to quantify via MDS the differences in the inter- and intra-molecular motions of these complexes, with a specific focus on what the authors describe as catch-bond behavior between the TCRs and pMHC, which could explain how T-cells can discriminate between different peptides in the presence of weak separating force.

Strengths:

The authors present extensive simulation data that indicates that, in both complexes, the number of high-occupancy inter-domain contacts initially increases with applied load, which is generally consistent with the authors' conclusion that both complexes exhibit catch-bond behavior, although to different extents. In this way, the paper expands our understanding of peptide discrimination by T-cells. The conclusions of the study are generally well supported by data. Further, the paper makes predictions about the relative strength of the catch-bond response of the two TCRs, which could be tested experimentally through protein mutagenesis and force application in Atomic Force Microscopy.

---

## [Referee Report · Reviewer #2 (Public review)]

In this work, Chang-Gonzalez and coworkers follow up on an earlier study on the force-dependence of peptide recognition by a T-cell receptor using all-atom molecular dynamics simulations. In this study, they compare the results of pulling on a TCR-pMHC complex between two different TCRs with the same peptide. A goal of the paper is to determine whether the newly studied B7 TCR has the same load-dependent behavior mechanism shown in the earlier study for A6 TCR. The primary result is that while the unloaded interaction strength is similar, A6 exhibits more force-stabilization.

This is a detailed study, and establishing the difference between these two systems with and without applied force may establish them as a good reference setup for others who want to study mechanobiological processes if the data were made available, and could give additional molecular details for T-Cell-specialists.

---

## [Referee Report · Reviewer #3 (Public review)]

Summary:

The paper by Chang-Gonzalez et al. is a molecular dynamics (MD) simulation study of the dynamic recognition (load-induced catch bond) by the T cell receptor (TCR) of the complex of peptide antigen (p) and the major histocompatibility complex (pMHC) protein. The methods and simulation protocols are essentially identical as those employed in a previous study by the same group (Chang-Gonzalez et al., eLife 2024). In the current manuscript the authors compare the binding of the same pMHC complex to two different TCRs, B7 and A6 which was investigated in the previous paper. While the binding is more stable for both TCRs under load (of about 10-15 pN) than in the absence of load, the main difference is that B7 shows a smaller amount of stable contacts with the pMHC than A6.

Strengths:

The topic is interesting because of the relevance of mechanosensing in biological processes including cellular immunology. The MD simulations provide strong evidence that different TCRs can respond mechanically in a different way upon binding the same pMHC complex. These findings are useful for interpreting how mechanical force is employed for modulating different function of T cells.

---

## [Author Response]

The following is the authors’ response to the original reviews.

**Reviewer 1:**
Summary:This paper describes molecular dynamics simulations (MDS) of the dynamics of two T-cell receptors (TCRs) bound to the same major histocompatibility complex molecule loaded with the same peptide (pMHC). The two TCRs (A6 and B7) bind to the pMHC with similar affinity and kinetics, but employ different residue contacts. The main purpose of the study is to quantify via MDS the differences in the inter- and intra-molecular motions of these complexes, with a specific focus on what the authors describe as catch-bond behavior between the TCRs and pMHC, which could explain how T-cells can discriminate between different peptides in the presence of weak separating force.Strengths:The authors present extensive simulation data that indicates that, in both complexes, the number of high-occupancy interdomain contacts initially increases with applied load, which is generally consistent with the authors’ conclusion that both complexes exhibit catch-bond behavior, although to different extents. In this way, the paper somewhat expands our understanding of peptide discrimination by T-cells.

a. The reviewer makes thoughtful assessment of our manuscript. While our manuscript is meant to be a “short” contribution, our significant new finding is that even for TCRs targeting the same pMHC, having similar structures, and leading to similar functional outcomes in conventional assays, their response to applied load can be different. This supports out recent experimental work where TCRs targeting the same pMHC differed in their catch bond characteristics, and importantly, in their response to limiting copy numbers of pMHCs on the antigen-presenting cell (Akitsu *et al.*, Sci. Adv., 2024).

Weaknesses:While generally well supported by data, the conclusions would nevertheless benefit from a more concise presentation of information in the figures, as well as from suggesting experimentally testable predictions.

b. We have updated all figures for clear and streamlined presentation. We have also created four figure supplements to cover more details.

Regarding testable predictions, an important prediction is that B7 TCR would exhibit a weaker catch bond behavior than A6 (line 297–298). This is a nontrivial prediction because the two TCRs targeting the same pMHC have similar structures and are functionally similar in conventional assays. This prediction can be tested by singlemolecule optical tweezers experiments. Based on our recent experiments Akitsu *et al.*, Sci. Adv. (2024), we also predict that A6 and B7 TCRs will differ in their ability to respond to cases when the number of pMHC molecules presented are limited. Details of how they would differ require further investigation, which is beyond the scope of the present work (line 314-319).

Another testable prediction for the conservation of the basic allostery mechanism is to test the C*β* FG-loop deletion mutant located at the hinge region of the *β* chain, where the deletion severely impairs the catch bond formation (line 261–264).

**Reviewer 2:**
In this work, Chang-Gonzalez and coworkers follow up on an earlier study on the force-dependence of peptide recognition by a T-cell receptor using all-atom molecular dynamics simulations. In this study, they compare the results of pulling on a TCR-pMHC complex between two different TCRs with the same peptide. A goal of the paper is to determine whether the newly studied B7 TCR has the same load-dependent behavior mechanism shown in the earlier study for A6 TCR. The primary result is that while the unloaded interaction strength is similar, A6 exhibits more force stabilization.This is a detailed study, and establishing the difference between these two systems with and without applied force may establish them as a good reference setup for others who want to study mechanobiological processes if the data were made available, and could give additional molecular details for T-Cell-specialists. As written, the paper contains an overwhelming amount of details and it is difficult (for me) to ascertain which parts to focus on and which results point to the overall take-away messages they wish to convey.

R2-a. As mentioned above and as the reviewer correctly pointed out, the condensed appearance of this manuscript arose largely because we intended it to be a Research Advances article as a short follow up study of our previous paper on A6 TCR published in eLife. Most of the analysis scripts for the A6 TCR study are already available on Github. For the present manuscript, we have created a separate Github repository containing sample simulation systems and scripts for the B7 TCR.

Regarding the focus issue, it is in part due to the complex nature of the problem, which required simulations under different conditions and multi-faceted analyses. We believe the extensive updates to the figures and texts make clearer and improved presentation. But we note that even in the earlier version, the reviewer pointed out the main take-away message well: “The primary result is that while the unloaded interaction strength is similar, A6 exhibits more force stabilization.

Detailed comments:(1) In Table 1 - are the values of the extension column the deviation from the average length at zero force (that is what I would term extension) or is it the distance between anchor points which is what I would assume based on the large values. If the latter, I suggest changing the heading, and then also reporting the average extension with an asterisk indicating no extensional restraints were applied for B7-0, or just listing 0 load in the load column. Standard deviation in this value can also be reported. If it is an extension as I would define it, then I think B7-0 should indicate extension = 0+/- something. The distance between anchor points could also be labeled in Figure 1A.

R2-b. “Extension” is the distance between anchor points that the reviewer is referring to (blue spheres at the ends of the added strands in Figure 1A). While its meaning should be clear in the section “Laddered extensions” in “MD simulation protocol” (line 357–390), in a strict sense, we agree that using it for the end-to-end distance can be confusing. However, since we have already used it in our previous two papers (Hwang *et al.*, PNAS 2020 and Chang-Gonzalez *et al.*, eLife, 2024), we prefer to keep it for consistency. Instead, in the caption of Table 1, we explained its meaning, and also explicitly labeled it in Figure 1A, as the reviewer suggested.

Please also note that the no-load case B7^0^ was performed by separately building a TCR-pMHC complex without added linkers (line 352), and holding the distal part of pMHC (the *α*3 domain) with weak harmonic restraints (line 406–408). Thus, no extension can be assigned to B7^0^. We added a brief explanation about holding the MHC *α*3 domain for B7^0^ in line 83–85.

(2) As in the previous paper, the authors apply ”constant force” by scanning to find a particular bond distance at which a desired force is selected, rather than simply applying a constant force. I find this approach less desirable unless there is experimental evidence suggesting the pMHC and TCR were forced to be a particular distance apart when forces are applied. It is relatively trivial to apply constant forces, so in general, I would suggest this would have been a reasonable comparison. Line 243-245 speculates that there is a difference in catch bonding behavior that could be inferred because lower force occurs at larger extensions, but I do not believe this hypothesis can be fully justified and could be due to other differences in the complex.

R2-c. There is indeed experimental evidence that the TCR-pMHC complex operates under constant separation. The spacing between a T-cell and an antigen-presenting cell is maintained by adhesion molecules such as the CD2CD58 pair, as explained in our paper on the A6 TCR Chang-Gonzalez *et al.*, eLife, 2024 and also in our previous review paper Reinherz *et al.*, PNAS, 2023. In in vitro single-molecule experiments, pulling to a fixed separation and holding is also commonly done. We added an explanation about this in line 79–83 of the manuscript. On the other hand, force between a T cell and antigen-presenting cell is also controlled by the actin cytoskeleton, which make the applied load not a simple function of the separation between the two cells. An explanation about this was added in line 300–303. Detailed comparison between constant extension vs. constant force simulations is definitely a subject of our future study.

Regarding line 243–245 of the original submission (line 297–298 of the revised manuscript), we agree with the reviewer that without further tests, lower forces at larger extensions *per se* cannot be an indicator that B7 forms a weaker catch bond. But with additional information, one can see it does have relevance to the catch bond strength. In addition to fewer TCR-pMHC contacts (Figure 1C of our manuscript), the intra-TCR contacts are also reduced compared to those of A6 (bottom panel of Figure 1D vs. Chang-Gonzalez *et al.*, eLife, 2024, Figure 8A,B, first column). Based on these data, we calculated the average total intra-TCR contact occupancies in the 500–1000-ns interval, which was 30.4±0.49 (average±std) for B7 and 38.7±0.87 for A6. This result shows that the B7 TCR forms a looser complex with pMHC compared to A6. Also, B7^low^ and B7^high^ differ in extension by 16.3 Å while A6^low^ and A6^high^ differ by 5.1 Å, for similar ∼5-pN difference between low- and high-load cases. With the higher compliance of B7, it would be more difficult to achieve load-induced stabilization of the TCR-pMHC interface, hence a weaker catch bond. We explained this in line 129–132 and line 292–297.

(3) On a related note, the authors do not refer to or consider other works using MD to study force-stabilized interactions (e.g. for catch bonding systems), e.g. these cases where constant force is applied and enhanced sampling techniques are used to assess the impact of that applied force: https://www.cell.com/biophysj/fulltext/S0006-3495(23)00341-7, https://www.biorxiv.org/content/10.1101/2024.10.10.617580v1. I was also surprised not to see this paper on catch bonding in pMHC-TCR referred to, which also includes some MD simulations: https://www.nature.com/articles/s41467-023-38267-1

R2-d. We thank the reviewer for bringing the three papers to our attention, which are:

(1) Languin-Catto¨en, Sterpone, and Stirnemann, Biophys. J. 122:2744 (2023): About bacterial adhesion protein FimH.

(2) Pen˜a Ccoa, *et al.*, bioRxiv (2024): About actin binding protein vinculin.

(3) Choi *et al.*, Nat. Comm. 14:2616 (2023): About a mathematical model of the TCR catch bond.

Catch bond mechanisms of FimH and vinculin are different from that of TCR in that FimH and vinculin have relatively well-defined weak- and strong-binding states where there are corresponding crystal structures. Availability of the end-state structures permits simulation approaches such as enhanced sampling of individual states and studying the transition between the two states. In contrast, TCR does not have any structurally well-defined weak- or strong-binding states, which requires a different approach. As demonstrated in our current manuscript as well as in our previous two papers (Hwang *et al.*, PNAS 2020 and Chang-Gonzalez *et al.*, eLife, 2024), our microsecond-long simulations of the complex under realistic pN-level loads and a combination of analysis methods are effective for elucidating the catch bond mechanism of TCR. These are explained in line 227–238 of the manuscript.

The third paper (Choi, *et al.*, 2023) proposes a mathematical model to analyze extensive sets of data, and also perform new experiments and additional simulations. Of note, their model assumptions are based mainly on the steered MD (SMD) simulation in their previous paper (Wu, *et al.*, Mol. Cell. 73:1015, 2019). In their model, formation of a catch bond (called catch-slip bond in Choi’s paper) requires partial unfolding of MHC and tilting of the TCR-pMHC interface. Our mechanism does not conflict with their assumptions since the complex in the fully folded state should first bear load in a ligand-dependent manner in order to allow any larger-scale changes. This is explained in line 239–243.

For the revised text mentioned above (line 227–243), in addition to the 3 papers that the reviewer pointed out, we cited the following papers:

• Thomas, *et al.,* Annu. Rev. Biophys. 2008: Catch bond mechanisms in general.

• Bakolitsa *et al.,* Cell 1999, Le Trong *et al.,* Cell 2010, Sauer *et al.,* Nat. Comm. 2016, Mei *et al.,* eLife 2020:

Crystal structures of FimH and vinculin in different states.

• Wu, *et al.*, Mol. Cell. 73:1015, 2019: The SMD simulation paper mentioned above.

(4) The authors should make at least the input files for their system available in a public place (github, zenodo) so that the systems are a more useful reference system as mentioned above. The authors do not have a data availability statement, which I believe is required.

R2-d. As mentioned in R2-a above, we have added a Github repository containing sample simulation systems and scripts for the B7 TCR.

**Reviewer 3:**
Summary:The paper by Chang-Gonzalez et al. is a molecular dynamics (MD) simulation study of the dynamic recognition (load-induced catch bond) by the T cell receptor (TCR) of the complex of peptide antigen (p) and the major histocompatibility complex (pMHC) protein. The methods and simulation protocols are essentially identical to those employed in a previous study by the same group (Chang-Gonzalez et al., eLife 2024). In the current manuscript, the authors compare the binding of the same pMHC to two different TCRs, B7 and A6 which was investigated in the previous paper. While the binding is more stable for both TCRs under load (of about 10-15 pN) than in the absence of load, the main difference is that, with the current MD sampling, B7 shows a smaller amount of stable contacts with the pMHC than A6.Strengths:The topic is interesting because of the (potential) relevance of mechanosensing in biological processes including cellular immunology.Weaknesses:The study is incomplete because the claims are based on a single 1000-ns simulation at each value of the load and thus some of the results might be marred by insufficient sampling, i.e., statistical error. After the first 600 ns, the higher load of B7^high^ than B7^low^ is due mainly to the simulation segment from about 900 ns to 1000 ns (Figure 1D). Thus, the difference in the average value of the load is within their standard deviation (9 +/- 4 pN for B7^low^ and 14.5 +/- 7.2 for B7^high^, Table 1). Even more strikingly, Figure 3E shows a lack of convergence in the time series of the distance between the V-module and pMHC, particularly for B7^0^ (left panel, yellow) and B7^low^ (right panel, orange). More and longer simulations are required to obtain a statistically relevant sampling of the relative position and orientation of the V-module and pMHC.

R3-a. The reviewer uses data points during the last 100 ns to raise an issue with sampling. But since we are using realistic pN range forces, force fluctuates more slowly. In fact, in our simulation of B7^high^, while the force peaks near 35 pN at 500 ns (Figure 1D of our manuscript), the interfacial contacts show no noticeable changes around 500 ns (Figure 2B and Figure 2–figure supplement 1C of our manuscript). Similarly slow fluctuation of force was also observed for A6 TCR (Figure 8 of Chang-Gonzalez *et al.*, eLife (2024)). Thus, a wider time window must be considered rather than focusing on forces in the last 100-ns interval.

To compare fluctuation in forces, we added Figure 1–figure supplement 2, which is based on Appendix 3–Figure 1 of our A6 paper. It shows the standard deviation in force versus the average force during 500–1000 ns interval for various simulations in both A6 (open black circles) and B7 (red squares) systems. Except for Y8A^low^ and dFG^low^ of A6 (explained below), the data points lie on nearly a straight line.

Thermodynamically, the force and position of the restraint (blue spheres in Figure 1A of our manuscript) form a pair of generalized force and the corresponding spatial variable in equilibrium at temperature 300 K, which is akin to the pressure *P* and volume *V* of an ideal gas. If *V* is fixed, *P* fluctuates. Denoting the average and std of pressure as ⟨*P*⟩ and ∆*P*, respectively, Burgess showed that ∆*P/*⟨*P*⟩ is a constant (Eq. 5 of Burgess, Phys. Lett. A, 44:37; 1973). In the case of the TCR*αβ*-pMHC system, although individual atoms are not ideal gases, since their motion leads to the fluctuation in force on the restraints, the situation is analogous to the case where pressure arises from individual ideal gas molecules hitting the confining wall as the restraint. Thus, the near-linear behavior in the figure above is a consequence of the system being many-bodied and at constant temperature. The linearity is also an indicator that sampling of force was reasonable in the 500–1000-ns interval. The fact that A6 and B7 data show a common linear profile further demonstrates the consistency in our force measurement. About the two outliers of A6, Y8A^low^ is for an antagonist peptide and dFG^low^ is the C*β* FG-loop deletion mutant. Both cases had reduced numbers of contacts with pMHC, which likely caused a wider conformational motion, hence greater fluctuation in force.

Upon suggestion by the reviewer, we extended the simulations of B7^0^, B7^low^ and B7^high^ to about 1500 ns (Table 1). While B7^0^ and B7^low^ behaved similarly, B7^high^ started to lose contacts at around 1300 ns (top panel of Figure 1D and Figure 2B). A closer inspection revealed that destabilization occurred when the complex reached low-force states. Even before 1300 ns, at about 750 ns, the force on B7^high^ drops below 5 pN, and another drop in force occurred at around 1250 ns, though to a lesser extent (Figure 1D). These changes are followed by increase in the Hamming distance (Figure 2B). Thus, in B7^high^, destabilization is caused not by a high force, but by a lack of force, which is consistent with the overarching theme of our work, the load-induced stabilization of the TCR*αβ*-pMHC complex.

The destabilization of B7^high^ during our simulation is a combined effect of its overall weaker interface compared to A6 (despite having comparable number of contacts in crystal structures; line 265–269), and its high compliance (explained in the second paragraph of our response R2-c above). Under a fixed extension, the higher compliance of the complex can reach a low-force state where breakage of contacts can happen. In reality, with an approximately constant spacing between a T cell and an antigen-presenting cell, force is also regulated by the actin cytoskeleton (explained in the first paragraph of R2-c above). While detailed comparison between constant-extension and constant-force simulation is the subject of a future study, for this manuscript, we used the 500–1000-ns interval for calculating time-averaged quantities, for consistency across different simulations. For time-dependent behaviors, we showed the full simulation trajectories, which are Figure 1D, Figure 2B, Figure 2–figure supplement 1 (except for panel E), and Figure 4–figure supplement 1B.

Thus, rather than performing replicate simulations, we perform multiple simulations under different conditions and analyze them from different angles to obtain a consistent picture. If one were interested in quantitative details under a given condition, e.g., dynamics of contacts for a given extension or the time when destabilization occurs at a given force, replicate simulations would be necessary. However, our main conclusions such as load-induced stabilization of the interface through the asymmetric motion, and B7 forming a weaker complex compared to A6, can be drawn from our extensive analysis across multiple simulations. Please also note that reviewer 1 mentioned that our conclusions are “generally well supported by data.”

A similar argument applies to Figure 2–figure supplement 1F (old Figure 3B that the reviewer pointed out). If precise values of the V-module to pMHC distance were needed, replicate simulations would be necessary, however, the figure demonstrates that B7^high^ maintains more stable interface before the disruption at 1300 ns compared to B7^low^, which is consistent with all other measures of interfacial stability we used. The above points are explained throughout our updated manuscript, including

• Line 106–110, 125–132, 156–158, 298–303.

• Figures showing time-dependent behaviors have been updated and Figure 1–figure supplement 2 has been added, as explained above.

It is not clear why ”a 10 A distance restraint between alphaT218 and betaA259 was applied” (section MD simulation protocol, page 9).

R3-b. *α*T218 and _β_A259 are the residues attached to a leucine-zipper handle in in vitro optical trap experiments (Das, *et al.*, PNAS 2015). In T cells, those residues also connect to transmembrane helices. Our newly added Figure 1–figure supplement 1 shows a model of N15 TCR used in experiments in Das’ paper, constructed based on PDB 1NFD. Blue spheres represent C*α* atoms corresponding to *α*T218 and *β*A259 of B7 TCR. Their distance is 6.7 Å. The 10- Å distance restraint in simulation was applied to mimic the presence of the leucine zipper that prevents excessive separation of the added strands. The distance restraint is a flatbottom harmonic potential which is activated only when the distance between the two atoms exceeds 10 Å, which we did not clarify in our original manuscript. It is now explained in line 371–373. The same restraint was used in our previous studies on JM22 and A6 TCRs.

**Recommendations for the authors:**

**Reviewer #1 (Recommendations for the authors):**
(1) Clarify the reason for including arguably non-physiological simulations, in which the C domain is missing. Is the overall point that it is essential for proper peptide discrimination?

R1-c. This is somewhat a philosophical question. Rather than recapitulating experiment, we believe the goal of simulation is to gain insight. Hence, a model should be justified by its utility rather than its direct physiological relevance. The system lacking the C-module is useful since it informs about the allosteric role of the C-module by comparing its behavior with that of the full TCR*αβ*-pMHC complex. The increased interfacial stability of V*αβ*-pMHC is also consistent with our discovery that the C-module likely undergoes a partial unfolding to an extended state, where the bond lifetime increases (Das, *et al.*, PNAS 2015; Akitsu *et al.*, Sci. Adv., 2024). In this sense, V*αβ*-pMHC has a more direct physiological relevance. Furthermore, considering single-chain versions of an antibody lacking the C-module (scFv) are in widespread use (Ahmad *et al.,* J. Immunol. Res., 2012) including CAR T cells, a better understanding of a TCR lacking the C-module may help with developing a novel TCR-based immunotherapy. These explanations have been added in line 253–261.

(2) Suggest changing V*αβ*-pMHC to B7^0^∆C to emphasize that the constant domain is deleted.

R1-d. While we appreciate the reviewer’s suggestion, the notation V*αβ*-pMHC was used in our previous two papers (Hwang, PNAS 2020, Chang-Gonzalez, eLife 2024). We thus prefer to keep the existing notation.

(3) Suggest adding A6 data to table 1 for comparison, making it clear if it is from a previous paper.

R1-e. Table 1 of the present manuscript and Table 1 of the A6 paper differ in items displayed. Instead of merging, we added the extension and force for A6 corresponding to B7^low^ and B7^high^ in the caption of Table 1.

(4) Suggest discussing the catch-bond behavior in terms of departure from equilibrium, e.g. is it possible to distinguish between different (catch vs slip) bond behaviors on the basis of work of separation histograms? If the difference does not show up in equilibrium work, the exponential work averages would be similar, but work histograms could be very different.

R1-f. Although energetics of the catch versus slip bond will provide additional insight, it is beyond the scope of the present simulations that do not involve dissociation events nor simulations of slip-bond receptors. We instead briefly mention the energetic aspect in terms of T-cell activation in line 316–319.

(5) Have the simulations in Figure 1 reached steady state? The force and occupancy increase almost linearly up until 500ns, then seem to decrease rather dramatically by 750ns. It might be worthwhile to extend one simulation to check.

R1-g. We did extend the simulation to about 1500 ns. The large and slow fluctuation in force is an inherent property of the system, as explained in R3-a above.

(6) Is the loss of contacts for B7^0^ due to thermalization and relaxation away from the X-ray structure?

R1-h. The initial thermalization at 300 K is not responsible for the loss of contacts for B7^0^ since we applied distance restraints to the initial contacts to keep them from breaking during the preparatory runs (line 358–370). While ‘relaxation away from the X-ray structure’ gives an impression that the complex approaches an equilibrium conformation in the absence of the crystallographic confinement, our simulation indicates that the stability of the complex depends on the applied load. We made the distinction between relaxation and the load-dependent stability clearer in line 233–238.

(7) Figure 4 contains a very large amount of data. Could it be simplified and partly moved to SI? For example, panel G is somewhat hard to read at this scale, and seems non-essential to the general reader.

R1-i. Upon the reviewer’s suggestion, we simplified Figure 4 by moving some of the panels to Figure 4–figure supplement 1. Panels have also been made larger for better readability.

(8) If the coupling between C and V domains is necessary for catch-bond behavior, can one propose mutations that would disrupt the interface to test by experiment? This would be interesting in light of the authors’ own comment on p. 8 that ’a logical evolutionary pressure would be for the C domains to maximize discriminatory power by adding instability to the TCR chassis,’ which might lead to a verifiable hypothesis.

R1-j. This has already been computationally and experimentally tested for other TCRs by the C*β* FG-loop deletion mutants that diminish the catch bond (Das, *et al.*, PNAS 2015; Hwang *et al.*, PNAS 2020; ChangGonzalez *et al.*, eLife, 2024). Furthermore, the V*γδ*-C*αβ* chimera where the C-module of TCR*γδ* is replaced by that of TCR_αβ_ that strengthens the V-C coupling achieved a gain-of-function catch bond character while the wild-type TCR*γδ* is a slip-bond receptor (Mallis, *et al.*, PNAS 2021; Bettencourt *et al.*, Biophys. J. 2024). We added our prediction that the FG-loop deletion mutants of B7 TCR will behave similarly in line 261–264.

(9) Regarding extending TCR and MHC termini using native sequences, as described in the methods, what would be the disadvantage of using the same sequence, which could be made much more rigid, e.g. a poly-Pro sequence? After all, the point seems to be applying a roughly constant force, but flexible/disordered linkers seem likely to increase force fluctuation.

R1-k. The purpose of adding linkers was to allow a certain degree of longitudinal and transverse motion as would occur in vivo. While it will be worthwhile to explore the effects of linker flexibility on the conformational dynamics of the complex, for the present study, we used the actual sequence for the linkers for those proteins (line 341–344).

**Reviewer #2 (Recommendations for the authors):**
(1) Figure 2 is almost illegible, especially Figure 2A-D. I do not think that these contacts vs time would be useful to anyone except for someone interested in this particular pMHC interaction, so I would suggest moving it to a supporting figure and making it much larger.

R2-e. Thanks for the suggestion. We created Figure 2–figure supplement 1 and made panels larger for clearer presentation.

(2) Figure 4 is overwhelming, and does not convey any particular message.

R2-f. This is the same comment as reviewer 1’s comment (7) above. Please see our response R1-i.

**Reviewer #3 (Recommendations for the authors):**
(1) The label ”beta2m” in Figure 1A should be moved closer to the beta2 microglobulin domain. A label TCR should be added to Figure 1A.

R3-c. Thanks for pointing out about *β*2m. We have corrected it. About putting the label ‘TCR,’ to avoid cluttering, we explained that V*α*, V*β*, C*α*, and C*β* are the 4 subdomains of TCR in the caption of Figure 1A.

(2) Hydrogen atoms should be removed from the peptide in Figure 1B.

R3-d. We have removed the hydrogen atoms.

(3) The authors should consider moving Figures 1 A-D to the SI and show a simpler description of the contact occupancy than the heat maps. The legend of Figure 2A-D is too small.

R3-e. By ‘Figures 1 A-D’ we believe the reviewer meant Figure 2A–D. This is the same comment as reviewer 2’s comment (1). Please see our response R2-e above.

(4) Vertical (dashed) lines should be added to Figure 3E at 500 ns to emphasize the segment of the time series used for the histograms.

R3-f. We added vertical lines in figures showing time-dependent behaviors, which are Figure 1D, Figure 2B, Figure 2–figure supplement 1F, and Figure 4–figure supplement 1B.